# Reproductive Morphology and Success in Annual versus Perennial Legumes: Evidence from *Astragalus* and the Fabeae (Papilionoideae)

**DOI:** 10.3390/plants13172380

**Published:** 2024-08-26

**Authors:** Andrey Sinjushin, Maria Ploshinskaya, Andrey Sytin

**Affiliations:** 1Legumes Department, Institute of Field and Vegetable Crops, 21101 Novi Sad, Serbia; 2Department of Higher Plants, Faculty of Biology, Lomonosov Moscow State University, 119234 Moscow, Russia; ploshinskaya@rambler.ru; 3Herbarium of Higher Plants, Komarov Botanical Institute of the Russian Academy of Sciences, 197022 Saint Petersburg, Russia; asytin@binran.ru

**Keywords:** evolution, flower, fruit productivity, inflorescence, lifespan, ovule, seed productivity

## Abstract

The third largest angiosperm family, Leguminosae, displays a broad range of reproductive strategies and has an exceptional practical value. Whereas annual legume species are mostly planted as crops, there is a significant interest in breeding and cultivating perennials. It is therefore of importance to compare reproductive traits, their interactions and the resulting productivity between related annual and perennial species. Two highly variable taxa were chosen for this purpose, the Fabeae tribe, including numerous temperate crops, and the largest angiosperm ‘megagenus’ *Astragalus*. A dataset of quantitative reproductive traits was composed of both originally obtained and previously published data. As a result of statistical analysis, we found that perennials in both groups tend to produce more flowers per axillary racemose inflorescence as well as more ovules per carpel. Perennial *Astragalus* also have larger flowers. Only a part of the developing flowers and ovules gives rise to mature pods and seeds. This difference is especially pronounced in small populations of rare and threatened perennials. Numerous reasons underlie the gap between potential and real productivity, which may be potentially bridged in optimal growing conditions.

## 1. Introduction

The evolution of reproductive traits is the key strategy of species’ adaptation to different environments. Not only is a reproductive strategy a matter of thriving, it also defines the practical value of a certain taxon as a crop or livestock. The diversity of modes of propagation often results in the broad distribution of a group with its representatives intensively used by humans. This is the case with many plant groups, including Leguminosae, the third largest angiosperm family, which also has the highest number of cultivated members [1].

Northern temperate latitudes are mostly inhabited by legumes belonging to the inverted repeat-lacking clade (IRLC) [2], a group of predominantly herbaceous plants. This clade is the most species-rich within the family and the vast majority of its diversity is attributed to the largest angiosperm ‘megagenus’ *Astragalus*, with ca. 3300 species. From a practical point of view, this genus seems underutilized, as only a small part of its species richness is used for applied, mostly medicinal, purposes [3]. Another diverse group of the IRLC is the Fabeae tribe, with ca. 400 spp., many of which possess outstanding value as the oldest known leguminous crops of subtropical and temperate latitudes, including pea (*Pisum sativum* L.), lentil (*Lens culinaris* Medik.), broad bean (*Vicia faba* L.), grass pea (*Lathyrus sativus* L.), and other valuable representatives of the large genera *Vicia* and *Lathyrus*.

Both *Astragalus* and the Fabeae include annual and perennial representatives. Numerous species of both groups are herbs, but *Astragalus* exhibits a much broader range of growth habits and general appearance. Shrubs and subshrubs, as well as stemless (acaulescent) rosette or cushion-shaped forms are present in many sections of the genus [4,5] but are not found in the Fabeae. The members of the latter often have a weak stem tethering to some support with leaf tendrils [6].

Despite pronounced differences in their growth forms and vegetative organs, two groups under discussion share numerous similarities in their reproductive structures. With rare exceptions, both *Astragalus* and the Fabeae produce flowers in axillary unbranched racemes or spikes, either bractless or with small bracts [7,8]. The number of flowers in each such inflorescence can range between one and several dozens. In the Fabeae, this raceme is usually one-sided [8] whereas in *Astragalus* flowers are arranged spirally around a raceme stalk. The length of a sterile portion (peduncle) of a raceme is very diverse, ranging from nearly absent (sessile or subsessile flowers) to very long, usually varying even within a single plant [4,8].

The overall floral morphology is quite similar between *Astragalus* and the Fabeae, viz. flowers that are hermaphroditic, monosymmetric, and nectariferous. Their corolla is papilionate, i.e., its petals are differentiated into three types: adaxial banner petal, lateral wings, and two abaxial petals fused to produce a keel. Typically, androecia of both taxa are diadelphous or pseudomonadelphous. Flowers of some *Astragalus* possess minute bracteoles [4].

Both autogamous and allogamous breeding systems are found in *Astragalus* (reviewed in [9]) and the Fabeae [10]. In each particular species or even population, degree of self-compatibility may vary from autogamy (sometimes in a form of cleistogamy) to obligate xenogamy [9,10,11].

Statistically, annuality is often associated with preferential self-pollination, although the exact evolutionary origin and significance of such a relation are debatable [12]. Such association exists at least in the Fabeae [10,11]. Historically self-pollination was interpreted as a derived trait [13]. However, the evolution of a lifespan is not unidirectional. In *Astragalus*, the perennial habit is an ancestral state with at least eight independent origins of annuals and two reversions to perenniality [14]. Oppositely, in the Fabeae the annual lifespan was reconstructed as ancestral with several shifts to perenniality and back again [15]. In both taxa, annuality is interpreted as an adaptation to the xeric climate of the Mediterranean, the area of origin of the Fabeae [15] and of the secondary diversification of *Astragalus,* initially emerging in West Asia [16].

Although the oldest known temperate legume crops are predominantly annuals, there is an increasing interest in shifting to breeding and cultivation of perennials which seem more promising in terms of sustainability and nutritional value [17] but their yield is often lower [18]. This raises questions as to whether the convergence in major patterns of reproductive biology exists among related species having different lifespans, as well as about the trade-offs among different components of productivity. This problem has an obvious importance for the conservation of rare species.

The opposite directions of the life cycle evolution make two related groups, *Astragalus* and members of the Fabeae, an intriguing model for the comparative study of the possible convergence of and correlation among reproductive features, with special reference to a lifespan. This work aims to investigate how values of different reproductive traits correlate with each other and how they may contribute to fruit and seed sets in annuals versus perennials.

## 2. Results

### 2.1. Quantitative Features of Reproductive Organs and Correlation between Them

Figure 1 compares values of selected reproductive parameters between annuals and perennials in two taxa. The following abbreviations are used hereafter: New World *Astragalus* = NWA, Old World *Astragalus* = OWA. A dataset used for these calculations is available in Appendix A.

In both the Fabeae and *Astragalus*, perennials have more flowers per inflorescence and more ovules per carpel (Figure 1B,H). The latter difference was not found significant in OWA, probably due to a very small number of annuals characterized in this group.

Perennial species of *Astragalus* possess larger flowers, as evidenced from calyx parameters (Figure 1C,E). This estimate is also reliable for corolla sizes, as banner and calyx lengths correlate strongly: in the pooled OWA and NWA sample, Pearson’s *r* = 0.85 (*p* < 0.01, n = 898). In the Fabeae, flower sizes differ insignificantly between annuals and perennials regardless of which parameter was compared (Figure 1C,D).

Perennial representatives of *Astragalus* also have larger seeds than annual congeners (Figure 1G) whereas perennials of the Fabeae have significantly smaller seeds than annuals in terms of their mass (Figure 1F). This difference remains statistically reliable even if one excludes three outlying annual (and, notably, crop) species, *Vicia faba*, *V. narbonensis* L., and *Lathyrus sativus*. However, the opposite holds true in a dataset of 30 annual and 22 perennial species of *Vicia* characterized with seed length in [19]. In this sample, annuals possess significantly smaller seeds (Mann–Whitney test, *p* < 0.05).

The ranges of variation expressed as standard deviation are often different between annuals and perennials: usually larger samples have a broader standard deviation range (Figure 1).

All statistically reliable correlations among studied features were found positive (Table 1). For all three groups under discussion, only the correlations between flower number and peduncle length and between flower and seed sizes are uniform. The rest of the correlations are relatively weak and, in many cases, insignificant (Table 1). Only NWA exhibits reliable positive correlations among most of the characters studied. Interestingly, the opposite, though weak, correlations are observed between ovule number and seed size in NWA and the Fabeae. In the latter case, the *p*-value indicates the insignificance of the correlation observed (*p* = 0.08) but approaches the conventional threshold of *p* = 0.05.

Although some traits are independent in the whole dataset of *Astragalus*, they may be correlated at the level of certain sections (Figure 2). For example, flower number and size correlate positively in sect. *Leptocarpi* but negatively in sect. *Scytocarpi* (Figure 2A), both from North America.

### 2.2. The Real Productivity and Its Components

Both previously published and original data indicate that both seeds to ovules (S/O) and fruits to flowers (F/F) ratios are significantly higher in annual species of the Fabeae than in perennials (Figure 3B,D). By contrast, these ratios differ insignificantly between annuals and perennials in *Astragalus* (Figure 3B,D), although there is a marked deficiency of data on the productivity of annual *Astragalus*. Average F/F ratios are similar between annual *Astragalus* and the Fabeae, whereas average S/O ratio are close between perennial representatives of both taxa: 0.45 in the Fabeae, 0.44 in the whole dataset of *Astragalus*, and 0.47 in the least concern species of *Astragalus* (Appendix A).

Table 2 represents the correlations among the components of productivity in both taxa. Data were pooled from previously published and originally obtained results (Appendix A). Only statistically significant correlations are discussed hereafter. Both *Astragalus* and the Fabeae exhibit strong positive correlations between numbers of flowers and fruits produced by a single axillary raceme as well as between numbers of ovules and seeds produced by a flower.

Many of the correlations are uniform between *Astragalus* and the Fabeae (Table 2). Among those which differ between taxa, one may note that the F/F ratio is strongly positively correlated with a number of fruits in *Astragalus,* whereas a strong negative correlation exists between F/F ratio and flower number in the Fabeae (Table 2). Similarly, S/O ratio correlates positively with seed number in *Astragalus* and negatively with ovule number in the Fabeae. Seed sizes did not correlate significantly with other components of productivity except for S/O ratio in *Astragalus* (low negative correlation, Table 2).

### 2.3. Non-Random Seed Abortion

Species of the Fabeae exhibit non-random distribution of the probabilities of seed development in different positions within a pod (Figure 4). The highest probability of successful seed development was recorded either in a middle or in a subdistal portion of a pod. In 102 pods of *V. hirsuta* (L.) Gray, which normally has only two ovules per ovary, a ratio of 86 proximal to 92 distal seeds was observed, insignificantly different from 1:1 (chi square test: *χ*^2^ = 0.202), i.e., there is an equiprobable distribution.

It should also be noted that the highest probabilities of successful seed development achieved in optimal position within a pod were unequal between studied species of the Fabeae. In perennials this probability neared 50% (39.8% in *V. sepium* L., 60.0% in *L. pratensis* L.), whereas in annuals it was much higher, approaching 100% (93.0% in *V. tetrasperma* (L.) Schreb., 90.2% in *V. hirsuta*) (Figure 4).

## 3. Discussion

### 3.1. Variation and Correlation of Traits between Annuals and Perennials

First and foremost, the obvious limitations of this metastudy should be emphasized. Despite the large amount of data analyzed, these can be only cautiously treated in some cases. For example, seed mass and linear sizes are associated indirectly, especially in the case of larger seeds [20], so these values are not interchangeable, which in turn complicates the comparison between taxa (Figure 1F,G). The same partly holds true for different estimates of flower size (Figure 1C–E). The dataset of productivity of *Astragalus* was also somewhat unbalanced due to a prevalence of observations on endangered and rare species.

Some regularities of reproductive traits variation are shared between both taxa. Perennials tend to produce more flowers per single inflorescence (Figure 1B) and multiple flowers are usually borne on longer peduncles (Table 1, see also [8]). The inflorescences reduced to one or few flowers are often (sub)sessile in both taxa. Independently from (the Fabeae, Figure 1C,D) or correlated with (NWA, Table 1) flower size, perennials produce more ovules per carpel (Figure 1H). The latter phenomenon is particularly noteworthy in the Fabeae, having no reliable differences in floral sizes between annuals and perennials. These patterns are especially interesting given the opposite and often recurrent evolution of a lifespan in two compared taxa. The convergence in flower sizes between annual *Astragalus* contrasts with the striking diversity of their pods (e.g., [21]) whereas fruit morphology is quite uniform among the Fabeae members. A higher potential productivity and, in some cases, attractiveness of perennials (see below) seem a common rule, at least for legumes and probably for most angiosperms [22]. Taking into account the recurrent and independent emergence of similar phenotypes in the two highly polymorphic taxa under discussion, it is likely that different regulatory mechanisms are behind the observed differences, such as contrasting flower or seed sizes. This agrees with the existing knowledge on the complex control of these traits dissected in model species (e.g., [23]).

As demonstrated by Snell and Aarssen [22], a pollination strategy, i.e., selfing or outcrossing, is probably a better predictor for reproductive morphology than a lifespan. Among taxonomically related annuals with contrasting pollination habits, selfers were found to have smaller flowers and seeds than outcrossers, which is mainly explained by the time limitation and selection for more rapid reproduction [22]. Whereas pollination strategy was not scored in our study, selfers prevail among annuals in the Fabeae [10,11]. In this tribe, flower sizes do not differ significantly between annuals and perennials (Figure 1C,D; [24]) possibly indicating that visual attraction is not a decisive factor in pollination of the tribe members. In the Fabeae, seeds are slightly smaller in perennials even if one removes large-seeded crops from the analysis (Figure 1F). However, both regularities reported in [22] can be found in *Astragalus* (Figure 1C,E,G). The difference in flower sizes between annual and perennial *Astragalus* is especially prominent if expressed in terms of calyx length (Figure 1E). Indeed, long tubular calyces are most likely adapted for selection of efficient animal pollinators typically visiting flowers of *Astragalus* (reviewed in [9]).

It is therefore not unlikely that the real contrast in reproductive morphology is between self- and cross-pollinators in both taxa rather than between annuals and perennials. There is an obvious lack of data on pollination mode in annual *Astragalus* but some data allow one to suggest that annuals may be preferentially selfers in this genus [25]. The autofertility is a quantitative and measurable trait which exhibits a gradient between its extremes, corresponding to obligate autogamous and obligate xenogamous species. Real species, independently from their lifespan, display a varying degree of autofertility, sometimes even on an intraspecific level [9,11,25].

Annual members of the Fabeae possess slightly heavier seeds, whereas the seeds of annual *Astragalus* are shorter (Figure 1F,G). On the one hand, this can be partly explained by the inclusion of large-seeded crop annuals into the dataset, although the outlying values were eliminated from further statistical analysis. On the other hand, earlier works have reported that lifespan and seed sizes are not tightly interconnected [18,26], which partly agrees with our results. One might anticipate the size of seeds decreasing with their number increasing and these two parameters are indeed correlated negatively, though weakly and unreliably (Table 2). Much more prominent is the negative association between seed size and S/O ratio in *Astragalus*, i.e., the smaller seeds are, the more fully the potential productivity is realized (Table 2).

Considering the variation of traits and correlation between them, it is of interest to take into account the age of all three groups under comparison, viz. OWA, NWA, and the Fabeae. The crown age of the Fabeae is estimated as 23.00–16.00 Mya [15]. Though the crown age of *Astragalus* has been revisited several times, the most recent estimate is about 11.47 Mya, with the Neo-*Astragalus* clade emerging about 9.84 Mya [27]. This may underlie the fact that NWA retain some correlations missing in OWA, a more ancient group which underwent a broader diversification (Table 1). However, the difference in the significance of correlations in two groups of *Astragalus* may be at least partly due to different sample sizes (Table 1). The Fabeae, the oldest-diverged group, also exhibits a broad range of morphological variation which complicates the search for association between certain traits (Table 1).

Our results on the opposite correlations between sections of *Astragalus* (Figure 2) indicate that the phylogenetic constraints take place in the course of adaptation to different environments. As a result, different syndromes and associations between characters evolve. Hence, there is seemingly no coordination between some features at the level of a genus but there are some patterns common for a particular section. Future studies on interspecific variation in contrasting environments and of different genotypes may help to disentangle these effects.

### 3.2. Realization of the Potential Productivity Is Unequal between Taxa and Lifespans

It is of little surprise that the potential productivity is only partially realized (Figure 3B,D). The proportion of flowers which shed without producing pods as well as the frequency of non-productive ovules may differ among years or populations of the same species (e.g., [28] cited from [9]). Non-fruiting flowers hypothetically provide pollen and increase the overall attractiveness of an inflorescence, but in some seasons they may give rise to mature fruits [29]. Not only do some flowers or ovules fail to be pollinated or reach maturity but there are also some preanthetic losses. For example, some floral buds in multiflorous inflorescences of *Vicia* abort long before blooming [8].

Two principal hypotheses aim to explain the lower seed and fruit set compared with ovule and flower number especially noteworthy in perennials [30,31]. The first rests upon the assumption that perennials are more prone to inbreeding depression due to the overall higher heterozygosity. The latter phenomenon is related to a higher frequency of self-pollination in annuals, so their genomes are more readily free from meiotic and embryonic lethal mutations [30]. These mutations may impair gameto- and embryogenesis, partly explaining the preanthetic losses of potential productivity. For example, Kolyasnikova [32] has recorded numerous anomalies in the development of embryo sacs in perennial *Astragalus*, i.e., a certain fraction of ovules degenerated prior to pollination. The populations of perennials mainly adapted to cross-pollination may suffer from inbreeding depression more severely which may explain low seed and fruit set in small, isolated populations of endemic or endangered perennials [33], i.e., exactly those from which much of the analyzed data on perennial *Astragalus* originate (Appendix A). Both F/F and S/O ratios differ significantly between annuals and perennials of the Fabeae but not of *Astragalus* (Figure 3B,D) which most probably results from a sort of sampling bias.

The conclusion of Soltani et al. [9] that F/F and S/O ratios are higher in outcrossers than in selfers confirms the preceding reasoning. The comparison in [9] was conducted in a group of 19 perennials, 10 of which are classified as endangered by the authors of publications involved in the metastudy. The genetic load in a population of self-compatible endangered species may be highly deleterious, whereas outcrossers may be less susceptible to this impact.

The second mechanism underlying a lower productivity of perennials is their necessity to allocate resources between reproduction and the accumulation of storage substances, as well as between all flowers and developing fruits [18]. The removal of some flowers increases seed and fruit set, as demonstrated in [29]. One may expect that the significant differences in productivity of the same species between seasons can be attributed to this reason.

Additionally, the deficiency of pollinators and pollen may also explain the low productivity of outcrossers [29]. The most obvious evidence for non-genetic factors interfering with successful maturation of seeds is their non-random distribution in a pod [34,35,36]. Together the lack of nutrients supplied from a pod stalk and a limited ability of pollen tubes to reach deeper ovules narrow down the highest probability of seed maturation to either (sub)apical portion or the middle of a pod (Figure 4, see also Figure 3 in [36]). The non-random seed abortion is found even in annuals that have short, few-seeded pods (Figure 4E,F). Notably, annuals have a higher probability of seed development than perennials even in the optimal position in a pod (Figure 4A,C,E). This can be mostly attributed to genetic causes, such as deleterious or lethal mutations, which are more harmful in perennials experiencing inbreeding depression [9]. To overcome the aforementioned deleterious factors, many more flowers and ovules are initiated in perennials than the resulting number of pods and seeds. Both F/F and S/O ratios could be higher in favorable conditions.

When examining connections between different quantitative components of productivity (Table 2), the most surprising is that both F/F and S/O ratios correlate more significantly with indicators of real productivity (numbers of fruits and seeds, respectively) in *Astragalus*. The parameters of potential productivity (flower and ovule number) seem to barely impact on F/F and S/O ratios. The opposite relation is recorded in the Fabeae (Table 2). Otherwise stated, the resulting fruit and seed yield (ratios, not absolute values) of *Astragalus* spp. are not strongly dependent on the numbers of flowers per inflorescence and ovules per flower. This is somewhat counterintuitive, as, at least due to mathematical reasons, the opposite tendency might be expected, because the numbers of flowers and ovules are denominators of F/F and S/O ratios, respectively. The members of the Fabeae demonstrate tendencies which are closer to the expected ones (Table 2).

Two explanations of this phenomenon are possible. As many *Astragalus* spp. listed in the dataset are threatened species from isolated populations (Appendix A), their potential productivity might be realized to a relatively small extent and have little impact on real fruit and seed set. If this explanation is exhaustive, one may expect that, in a smaller dataset excluding *Astragalus* species of most concern, average values of F/F and S/O ratios would be higher than in a full dataset of this genus. However, this expectation is not met (Table 2). Except for the correlations in samples too small to be taken into account, narrowing down the analysis to *Astragalus* species of least concern does not invert the correlations but can make them less significant, as in the case of flower number and F/F ratio (Table 2).

An alternative hypothesis is connected with the unusual biseriate arrangement of ovules in ovaries of *Astragalus*, quite rare among legumes. The range of ovule numbers is exceptionally broad in this genus and some species are outstanding in having very high values, e.g., more than 80 ovules in *A. praelongus* E. Sheld. [4]. One might predict that the development of so many ovules and potential seeds requires an increase of pod length, which is, in its turn, associated with the more severe expression of non-random seed abortion. However, in each position along a pod length the probability of a successful seed maturation is doubled due to a biseriate placement of ovules [36]. As a result, more ovules can be fertilized and the impact of non-random seed abortion becomes less significant. As seedless leguminous pods usually do not maturate, the higher probability of the fertilization of at least some ovules increases the likelihood of pod development and hence the F/F ratio.

### 3.3. A Trade-Off between Allocated Resources and the Resulting Productivity

One may expect that there is a complex regulatory interaction between different components of productivity, allowing for the avoidance of the excessive expenses of flower/inflorescence development. Indeed, some phenomena that can be classified like this exist. For example, Kupicha [37] has pointed out that perennial species of *Vicia* with many-flowered inflorescences tend to have narrower banner petals, so that the whole inflorescence rather than a single flower serves as an attractive unit. As a result, the negative correlation between flower number and size is likely to be predicted. However, no such correlation is found in the data analyzed here (Table 1 and Table 2). This agrees with previous reports on particular genera [8], although such a trade-off is present at a macroevolutionary scale (see [38] and references cited therein). The negative relation between flower and ovule numbers is also not found (Table 1 and Table 2). Beyond expectations, seed size only weakly compensates for the number of seeds produced in a single pod, although there is a low negative correlation between seed size and S/O ratio in *Astragalus* (Table 2). Much more prominent is such a trade-off in crops [39,40]. The available data from model species suggest that seed number and size are separable traits [41], so a trade-off between them, if present, is most likely driven by natural selection rather than pleiotropy or linkage of regulatory loci.

A negative correlation between flower and seed number found in *Astragalus* (Table 2) probably reflects the only compensatory mechanism found in the analyzed data. Although such a pattern was not discovered in the Fabeae, a negative statistical association between flower number and S/O ratio in this tribe seems to result from the same adaptation.

## 4. Materials and Methods

### 4.1. Dataset

Both originally acquired and previously published data were used for this study. Data on quantitative features of North American species of *Astragalus* were mostly extracted from the excellent work of R. Barneby [4]. Features of the Old World *Astragalus* originate from the monograph of D. Podlech and S. Zarre [5], although not all sections listed in this book were included in statistical analysis. Both sources were complemented by data from papers published later than 2013 (preferentially those which reported ovule numbers, the least available parameter in literature) and the authors’ personal observations. The raw data, together with exact sources of each value, are available in Appendix A.

It is important to note that, in all cases where data were extracted from previously published works, these publications served as a source of taxonomic information, such as generic or section attribution, which may not coincide with more recent views.

In both major monographs [4,5], values of each quantitative feature are given as a min–max range. Therefore, mid-range, i.e., 0.5⋅(max + min), was used as a measure of a ‘typical’ value. To assess how mid-range values that are reported in the literature correlate with averages obtained by us for the same species, data for calyx length (9 species) and flower number per raceme (25 species) were analyzed. Spearman’s correlation coefficients comprised 0.98 and 0.72 (*p* < 0.01), respectively, suggesting that the mid-range value may serve as a reliable measure. Additionally, this inspires optimism that these values are at least partly reproducible among regions and environments.

For description of calyx diameter in the Fabeae, mostly the same specimens as listed in [24] were used, partly supplemented by some newly obtained data, totally comprising 58 observations of 44 species. The book of Stankevich and Repjev [19], mostly devoted to *Vicia* species, was also used as a source of data.

In all cases, productivity characteristics were assessed for a single axillary racemose inflorescence rather than a whole plant. This is connected with a pronounced variation of the number of such racemes depending on environmental conditions, the number of branches and other factors [7,36]. In perennials, the age of a plant also contributes to this variation. For example, 4- and 5-year-old plants of *A. propinquus* Schischk. were reported to produce 40 times more seeds than those of the second year [42]. That is why all parameters were analyzed only for the minimum inflorescence unit. On the other hand, such a unit is usually characterized in monographs, which makes data from different sources compatible. When data on productivity were reported for several populations of the same species in the same paper, an average value was used for further analysis.

It should be highlighted that of 126 observations of 85 species of *Astragalus* reviewed for this work (Appendix A), 50 reports are devoted to species referred to as rare, endangered or endemic, often included in regional Red Lists. Oppositely, data on productivity of representatives of the Fabeae are obtained for widely distributed, least concern species, some of which are cultivated. This suggests a risk of sampling bias, so the comparison between taxa should be carried out and interpreted with caution. That is why some of the analyses in *Astragalus* were conducted on a subset consisting of least concern species (76 observations, 49 species); all annual *Astragalus,* except for *A. lentiginosus* Douglas ex Hook., belong to this group.

Lifespan was encoded as a binary variable, i.e., either annual or perennial. If a literature source reported a species as annual or biennial, this was further treated as annual. Species described as biennials or perennials were referred to as perennials in further statistical analysis. A few species listed as ‘annual or perennial’ were not included in calculations.

### 4.2. Data Acquisition and Statistical Analysis

The original data on productivity were obtained in natural populations as well as on experimental plot at the Skadovskii Zvenigorod Biological Station (Moscow region, Russia). Localities of populations with geographical coordinates (where possible) are also available in Appendix A. The fruits/flowers (F/F) ratio, or fruit set, was evaluated on the fruiting stage. The number of flowers which shed before fruiting was estimated based on peduncle scars and, in some species, persistent bracts. The seeds/ovules (S/O) ratio, or seed set, was evaluated in (almost) mature pods. Underdeveloped seeds and abortive ovules were counted using a stereomicroscope. In some species of the Fabeae, positions of mature seeds were recorded within a pod to characterize non-random seed abortion. Samples of 100–110 pods were analyzed for this purpose. To describe patterns of non-random seed abortion, both absolute and relative positions of ovules were scored as described in [36].

In the Fabeae members, the seed mass was weighed with a Pioneer PA64 electronic scale (Ohaus, Parsippany, NJ, USA). To obtain 100 seed mass value, fewer seeds were weighed and then recalculated if small seed samples were available. For *Astragalus*, seed length obtained from literature [4,5] served as a measure of seed size.

All statistical procedures were carried out with usage of Statistica 12 (Statsoft, Tulsa, OK, USA) and Microsoft Excel (Microsoft, Redmond, WA, USA). When comparing between annuals and perennials, in many cases Levene’s test indicated the significant differences of variances. Moreover, distributions of many parameters were obviously right-skewed (Figure 1). Therefore, both the Welch’s *t*-test and the nonparametric Mann–Whitney test were applied. In none of the cases did the results of two tests contradict each other. Outlying and extreme values were not included in comparison.

## 5. Conclusions

Although the two compared taxa are exceptionally diverse and widespread, there is a clear lack of data on their productivity in natural populations, especially of annual and, surprisingly, of least concern species of *Astragalus*. The existing data on fruit and seed sets, as well as reproductive strategy, mostly originates from works inspired by nature conservation motives and focused on threatened and rare species.

Taking into account the opposite directions of life history evolution in the Fabeae and *Astragalus,* alongside numerous reversals, the associations found in both taxa may be evaluated as universal and constituting a sort of lifespan constraint. These are the development of more flowers per inflorescence and more ovules per carpel in perennials; those of *Astragalus* usually develop larger flowers. These adaptations, clearly convergent and most likely having various genetic and physiological mechanisms, seem to compensate for a lower number of seeds per pod and often lower fruits/flowers and seeds/ovules ratios in perennials. As the vast majority of legume crops are annuals with relatively high values of F/F and S/O ratios, there is a limited possibility to enhance their yield via the increase of these ratios, which is not the case of perennials. The exact reasons behind the substantial difference between potential and real productivity are manifold, connected with both gene load and resource availability. This gap, however, seems bridgeable in a favorable environment which makes the breeding of perennial crops a promising goal.

## Figures and Tables

**Figure 1 plants-13-02380-f001:**
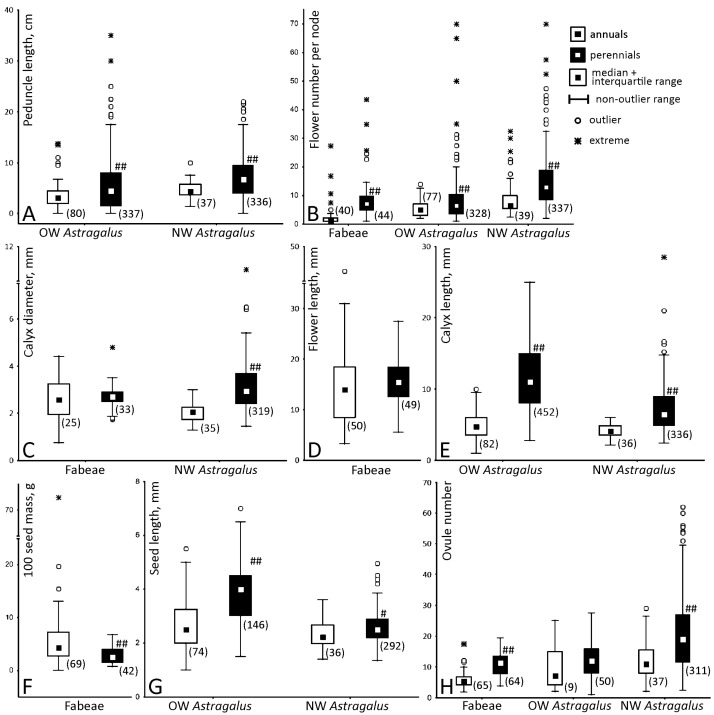
Variation of reproductive traits in the Fabeae and *Astragalus*. (**A**) Peduncle length, (**B**) the number of flowers per axillary inflorescence, (**C**–**E**) flower size expressed in terms of (**C**) calyx diameter for the Fabeae and NWA, (**D**) flower length for the Fabeae, (**E**) calyx length for *Astragalus*, (**F**–**G**) seed size expressed as (**F**) 100 seed mass for the Fabeae, (**G**) seed length for *Astragalus*, (**H**) ovule number. Sample size is given in parentheses. The Welch’s *t*-test was used for comparison between annuals and perennials; if at least one of the compared groups was small (n < 30), the Mann–Whitney test was applied. Significant difference is indicated with a hash sign, as follows: ^#^, *p* < 0.05, ^##^, *p* < 0.01.

**Figure 2 plants-13-02380-f002:**
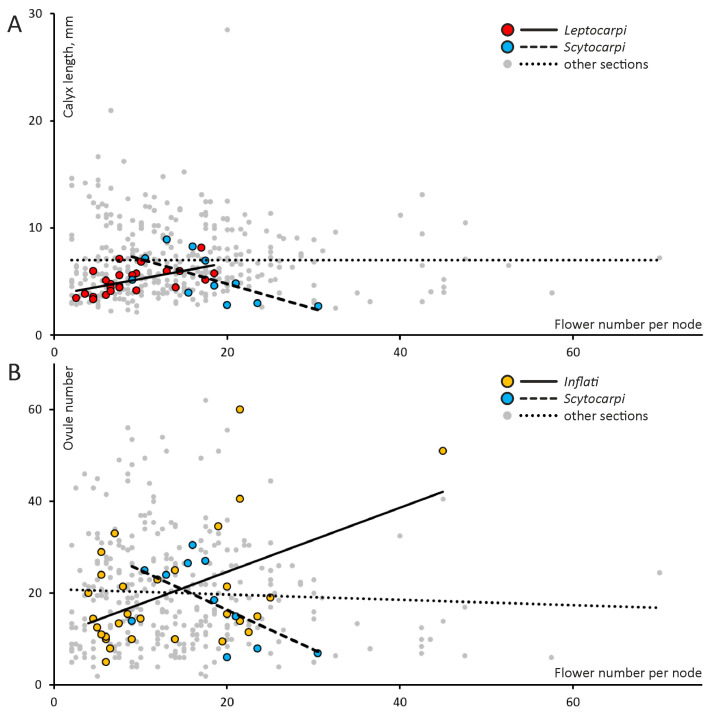
Opposite correlations in certain sections of North American *Astragalus*. (**A**) Flower number and calyx length correlate positively in sect. *Leptocarpi* (Spearman’s *ρ* = 0.63, *p* < 0.01, n = 24) and negatively in sect. *Scytocarpi* (*ρ* = −0.71, *p* < 0.05, n = 11). The remaining species exhibit no reliable correlation between these two parameters (Pearson’s *r* = −0.00, *p* = 1.00, n = 333). (**B**) Flower and ovule numbers exhibit a weak positive correlation in sect. *Inflati* (*ρ* = 0.18, *p* = 0.39, n = 30) and weak negative in sect. *Scytocarpi* (*ρ* = −0.52, *p* = 0.10, n = 11). In the remainder of the dataset, these traits are not correlated (*r* = −0.04, *p* = 0.53, n = 303).

**Figure 3 plants-13-02380-f003:**
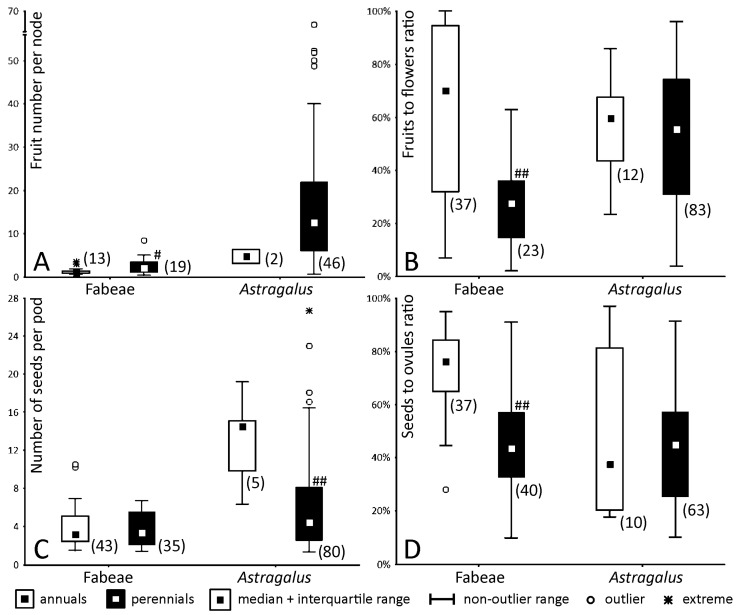
Variation of productivity features in selected taxa. (**A**) The number of fruits produced by a single axillary inflorescence, (**B**) fruits to flowers ratio, (**C**) the number of seeds per pod, and (**D**) seeds to ovules ratio. Sample size is given in parentheses. The Welch’s *t*-test was used for comparison between annuals and perennials; if at least one of the compared groups was small (n < 30), the Mann–Whitney test was applied. Due to the small number of annual *Astragalus* characterized with fruit number (**A**), no statistical comparison was conducted. Significant differences are indicated with hash signs: ^#^, *p* < 0.05, ^##^, *p* < 0.01.

**Figure 4 plants-13-02380-f004:**
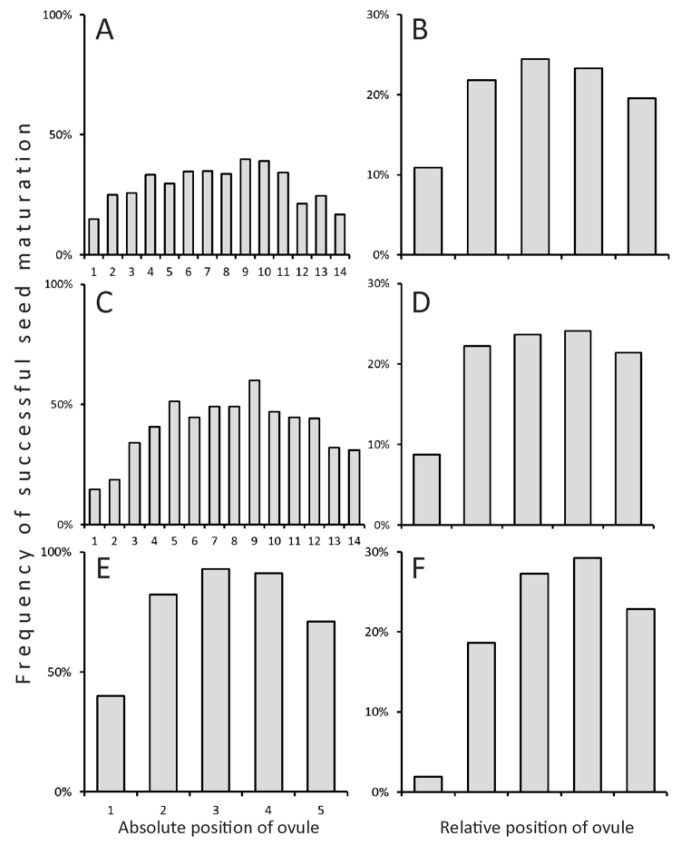
Patterns of non-random seed abortion in *Vicia sepium* (**A**,**B**), *Lathyrus pratensis* (**C**,**D**), and *V. tetrasperma* (**E**,**F**) represented as frequencies of seed maturation plotted against absolute (**A**,**C**,**E**) and relative (**B**,**D**,**F**) positions of ovule. The minimum of the horizontal axis corresponds to the proximal part of a pod.

**Table 1 plants-13-02380-t001:** Correlation (Spearman’s *ρ*) among the parameters of reproductive organs in a whole dataset of NA and OW *Astragalus* and the Fabeae (Appendix A). Numbers of analyzed observations are given in brackets.

		Flower Number	Peduncle Length	Flower Size ^a^	Ovule Number
**Peduncle length**	**OW**	**0.65 **** (282)			
	**NW**	**0.39 **** (367)			
	**F**	**0.76 **** (36)			
**Flower size ^a^**	**OW**	0.14 (403)	0.18 (389)		
	**NW**	0.01 (368)	**0.18 **** (364)		
	**F**	–0.03 (104)	0.09 (36)		
**Ovule number**	**OW**	–0.07 (50)	0.28 (48)	–0.01 (59)	
	**NW**	–0.00 (344)	**0.19 **** (340)	**0.66 **** (335)	
	**F**	0.03 (74)	–	–	
**Seed size ^b^**	**OW**	–0.05 (170)	–0.18 (185)	**0.79 **** (220)	–0.13 (24)
	**NW**	0.08 (324)	**0.24 **** (322)	**0.35 **** (315)	**0.35 **** (303)
	**F**	–0.10 (30)	–0.14 (21) ^c^	**0.53 **** (57) ^c^	–0.35 (63)
		–0.08 (55) ^c^			

F = Fabeae. Dash = no data. Significant correlations are marked with bold and asterisk: **, *p* < 0.01. ^a^ Different parameters were used for two taxa, viz. calyx length for *Astragalus* and flower length for the Fabeae (Appendix A). ^b^ Different parameters were used for two taxa, viz. seed length for *Astragalus* and 100 seed mass (g) for the Fabeae (Appendix A) except for ^c^. ^c^ These correlations are based on the data on seed length (Appendix A).

**Table 2 plants-13-02380-t002:** Correlation (Spearman’s *ρ*) among the parameters of productivity in the whole dataset of *Astragalus* (A, Appendix A), least concern species of *Astragalus* (LC), and the species of the Fabeae (F, Appendix A). Numbers of analyzed observations are given in brackets.

		Fruit Number	Flower Number	F/F Ratio	Seed Number	Ovule Number	S/O Ratio
**Flower number**	**A**	**0.83 **** (40)					
	**LC**	**0.80 **** (18)					
	**F**	**0.57 **** (32)					
**F/F ratio**	**A**	**0.74 **** (41)	**0.38 **** (52)				
	**LC**	**0.74 **** (18)	0.35 (29)				
	**F**	–0.11 (32)	**–0.81 **** (33)				
**Seed number**	**A**	**–0.48 **** (45)	**–0.45 **** (28)	–0.24 (64)			
	**LC**	–0.40 (22)	**–0.44 *** (27)	0.12 (32)			
	**F**	–0.30 (29)	–0.21 (32)	0.19 (33)			
**Ovule number**	**A**	–0.32 (25)	–0.31 (24)	–0.12 (41)	**0.64 **** (42)		
	**LC**	0.30 (5)	0.30 (5)	0.00 (16)	**0.58 *** (14)		
	**F**	0.12 (28)	0.33 (30)	**–0.46 **** (33)	**0.54 **** (72)		
**S/O ratio**	**A**	–0.34 (25)	–0.32 (24)	–0.05 (48)	**0.79 **** (50)	0.20 (45)	
	**LC**	**–0.90 *** (5)	**–0.90 *** (5)	0.42 (19)	**0.81 **** (20)	0.37 (17)	
	**F**	**–0.40 *** (29)	**–0.64 **** (31)	**0.60 **** (32)	0.11 (71)	**–0.67 **** (71)	
**Seed size ^a^**	**A**	0.11 (34)	0.07 (35)	0.10 (67)	–0.13 (58)	0.03 (30)	**–0.35 *** (46)
	**LC**	0.19 (20)	0.17 (21)	0.01 (45)	–0.12 (38)	0.02 (17)	**–0.40 *** (28)
	**F**	–0.27 (26)	–0.36 (30)	0.10 (40)	–0.12 (65)	–0.18 (63)	0.07 (63)

Significant correlations are marked with bold and asterisk: *, *p* < 0.05, **, *p* < 0.01. ^a^ Different parameters were used for two taxa, viz. seed length (mm) for *Astragalus* and 100 seed mass (g) for the Fabeae (Appendix A).

## Data Availability

Data are contained within the article and Appendix A.

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
