# Peer review of "Reproductive Morphology and Success in Annual versus Perennial Legumes: Evidence from Astragalus and the Fabeae (Papilionoideae)"

_plants, 2024, doi:10.3390/plants13172380_

Round 1

Reviewer 1 Report

Comments and Suggestions for Authors

In this article, Sinjushin and his colleagues aim to compare reproductive traits between related annual and perennial species, suggesting that perennials tend to produce more flowers per axillary racemose inflorescence as well as more ovules per carpel, and investigate how values of different reproductive traits correlate with each other and how they may contribute to fruit and seed set in annuals versus perennials. The manuscript enriches our knowledge about the reproductive strategy between the annual and perennial species of third largest angiosperm family, Leguminosae. The conclusions are supported by the data, and the he submitted manuscript is written well and interest to the readers. However, I have several comments that should be addressed before publication.

In scientific aspects:

1.       In both the Fabeae and Astragalus, perennials have more flowers per inflorescence and more ovules per carpel. Commonly, the size of the flower and seed will decrease when their number increased. However, Why the Perennial species of Astragalus possess larger flower sizes and larger seeds but the Perennial species of Fabeae possess insignificant flower sizes and the smaller seeds? A further extensive discussion in discussion section will enhance the readability and interesting of the manuscript.

Comments on the Quality of English Language

In language aspect:

1.  Make sure that the Latin names (genus name + specific epithet) of species should be in italic format (Page3, L118). Such as Vicia faba and V. narbonensis L., there are only the examples. Please check the whole manuscript and correct them.

2.  The authors must carefully check grammar, punctuation, spelling, and overall style in the whole text.

Author Response

Dear colleague,

Many thanks for your review of our manuscript and the suggestions made. Here is a brief outline of the corrections we introduced according to your comments. All changes are highlighted with red for your convenience in the manuscript file.

Q:  In both the Fabeae and Astragalus, perennials have more flowers per inflorescence and more ovules per carpel. Commonly, the size of the flower and seed will decrease when their number increased. However, Why the Perennial species of Astragalus possess larger flower sizes and larger seeds but the Perennial species of Fabeae possess insignificant flower sizes and the smaller seeds? A further extensive discussion in discussion section will enhance the readability and interesting of the manuscript.
A: Thank you for highlighting this interesting point. We added a brief discussion on the connection between seed number and size, see lines 220-221, 259-263. In lines 377-379, we also referenced a paper reporting that these parameters are separable on regulatory level, as seen from model plant species.

Q: Make sure that the Latin names (genus name + specific epithet) of species should be in italic format (Page3, L118). Such as Vicia faba and V. narbonensis L., there are only the examples. Please check the whole manuscript and correct them.
A: The original formatting was partly lost in the course of the editorial preparation for review. Anyway, we have restored italics in generic/specific names where required.

Q: The authors must carefully check grammar, punctuation, spelling, and overall style in the whole text.
A: Before resubmission, we checked all language aspects as well as style of the manuscript.

Thank you again for your helpful suggestions which have undoubtedly made our paper more readable.

Best wishes,
Authors 

Reviewer 2 Report

Comments and Suggestions for Authors

Very interesting work in eco-friendly plants.

Advice for authors:

Line 84….. between different components change in….. among different components

Lene 88 … between reproductive features change in among reproductive features

Line100   Astragalus change in Astragalus

Line 102  Astragalus change in Astragalus

Line 103  Astragalus change in Astragalus

Line 106 p< 0.05 and p<0.01 change in p< 0.05 and p<0.01

Line 107 Astragalus change in Astragalus

Line 112  p<0.01 change in p<0.01

Line 115 Astragalus change in Astragalus

Line 118 Vicia faba, V. narbonensis L change Vicia faba, V. narbonensis L

Line 119 Lathyrus sativus change in Lathyrus sativus

Line 120 Vicia change in Vicia

Line 121 , p < 0.05 change in , p < 0.05

Line 125 All statistically reliable correlations between ????

Line 128 … between most of the characters studied change in among most of the characters studied

Lines 131, 132…. p change in p

Line 133…. between parameters change in among parameters

Line 141 Astragalus change in Astragalus

Lines 147-148-149-150 151-152     p change in p

Line 167 …Astragalus change in Astragalus

Line 169 ….p change in p

Line 172   Astragalus change in Astragalus

Line 176….. between parameters change in among parameters

Line 184 and 186     Astragalus change in Astragalus

Line 193    V. hirsuta change in V. hirsuta

Line 264 …differ between years change in differ among years

Line 285 .. Soltani et al. change in Soltani et al.

Author Response

Dear colleague,

Thank you for your work on our manuscript and helpful comments. The changes in formatting you recommended has been introduced and highlighted with red alongside corrections suggested by other reviewers.
We did not manage to find any recommendations to italicize 'et al.', so, basing on our preceding experience of publishing in MDPI journals, we retained the original formatting. We hope this controversy can be solved in the course of the manuscript preparation for publication.

Best regards,
Authors 

Reviewer 3 Report

Comments and Suggestions for Authors

The study entitled ''Reproductive Morphology and Success in Annual versus Perennial Legumes: Evidence from Astragalus and the Fabeae (Papilionoideae)'' by Sinjushin et al. presented the comparison of reproductive traits between annual and perennial legume species within the Fabeae tribe and the Astragalus genus. The authors have utilized a dataset of quantitative reproductive traits from both original research and previously published data. The analysis reveals that perennials generally produce more flowers per inflorescence and ovules per carpel, with Astragalus perennials also having larger flowers. However, only a fraction of these flowers and ovules develop into mature pods and seeds, particularly in small populations of rare and threatened perennials. The study highlights the gap between potential and actual productivity, which could be mitigated under optimal growing conditions. This is an interesting study for future molecular breedings studies related to these plants, however, i have some minor concerns, which can be considered to strengthen the scientific cohernece of the manuscript. 

Comments of Authors:

1. In your study, seed size showed significant differences between annual and perennial species within Astragalus and the Fabeae. What do you hypothesize is the underlying genetic or developmental basis for these differences in seed size? Could this be linked to specific evolutionary pressures or constraints associated with the life cycle of annuals versus perennials?

2.  Given the diversity within Astragalus and the Fabeae, how do you account for phylogenetic relatedness when comparing reproductive traits? Could the observed differences be partially driven by phylogenetic inertia rather than adaptive evolution? How might you disentangle these effects in future studies?

3. How do environmental factors such as soil fertility, water availability, and climate influence the reproductive traits observed in annual versus perennial species in your study? Have you considered the potential role of plasticity in these traits, and how might this be integrated into your analyses?

4. Your study suggests trade-offs between different reproductive traits, such as seed size and number. How do you propose that these trade-offs are mediated at the physiological or molecular level? Are there known regulatory pathways or hormonal signals that might explain these patterns in legumes?

5. Given the differences in reproductive traits between annual and perennial species, how might these findings inform breeding strategies for improving yield or stress resilience in legumes? Are there specific traits identified in your study that could be targeted for selection in breeding programs? If yes, it should be discussed properly.

6. The seed-to-ovule and fruit-to-flower ratios differ between annual and perennial species. How do you interpret these differences in the context of reproductive success and fitness? Are these ratios consistent across different environments, or do they vary with ecological conditions?

7.  The study notes a correlation between flower size and reproductive success. How might flower size influence pollinator behavior and effectiveness in annual versus perennial species? Could differences in pollinator interactions contribute to the observed patterns in reproductive traits?

8. Are there other traits, such as seed dormancy, germination timing, or root architecture, that you think could be important in understanding the life history strategies of annual versus perennial legumes?

9. The study mentions a positive correlation between flower and seed sizes in Astragalus. Do you think this correlation is driven by genetic linkage, pleiotropy, or adaptive evolution? How would you design an experiment to test the causal relationship between these traits?

Comments on the Quality of English Language

Minor editing of English language required.

Author Response

Dear colleague,

First of all, we would like to express our gratitude to you for your thorough review of our manuscript and challenging questions. We responded to them (see below) and, inspired by some of them, added some pieces to the discussion. For your convenience, all changes compared with the initial submission were highlighted with red in the manuscript.

Q:  In your study, seed size showed significant differences between annual and perennial species within Astragalus and the Fabeae. What do you hypothesize is the underlying genetic or developmental basis for these differences in seed size? Could this be linked to specific evolutionary pressures or constraints associated with the life cycle of annuals versus perennials?
A: The genetics and developmental bases, not alternatives but more likely partly overlapping categories, were beyond the scope of this paper. As we were discussing very large taxa where similar phenotypes emerged recurrently and independently, one may suggest different regulatory mechanisms behind these differences. It agrees with the observations from model plant species such as crops. We opt for the existence of some constraints imposed by a life cycle but this, in its turn, is a product of evolutionary pressure. These two categories are also difficult to separate. Some discussion was added to the text (lines 227-231).

Q: Given the diversity within Astragalus and the Fabeae, how do you account for phylogenetic relatedness when comparing reproductive traits? Could the observed differences be partially driven by phylogenetic inertia rather than adaptive evolution? How might you disentangle these effects in future studies?
A: Our study is operating by large categories, sometimes not fully corresponding to taxonomy, such as Old World Astragalus, New World Astragalus and the Fabeae. Our results on the opposite correlations between sections of Astragalus (see Figure 2) indicate that the phylogenetic inertia does take place in the course of adaptation to different environments. As a result, different syndromes and associations between characters evolve. Hence, there are seemingly no coordination between some features on the level of a genus but there are some patterns common for a particular section. Future studies on interspecific variation in contrasting environments and of different genotypes may help to disentangle these effects. This discussion has been added (lines 274-280).

Q: How do environmental factors such as soil fertility, water availability, and climate influence the reproductive traits observed in annual versus perennial species in your study? Have you considered the potential role of plasticity in these traits, and how might this be integrated into your analyses?
A: These factors primarily influence the resulting productivity of a plant (such as the number of pods and seeds produced) – but not only. This work bases on the analysis of a dataset pooled from different previously published works devoted to numerous species in different environments and conditions. These works, in their turn, often refer to dozens of herbarium specimens collected in different years and sites. It is hardly possible to reproduce all or even a part of these hundreds of species in the same controlled conditions. However, the acceptable level of correlation between the data obtained by us and found in earlier works (see Materials and Methods) inspires some optimism that these values are reproducible among regions and environments. This note has been added to the Materials and Methods section (lines 405-306).

Q: Your study suggests trade-offs between different reproductive traits, such as seed size and number. How do you propose that these trade-offs are mediated at the physiological or molecular level? Are there known regulatory pathways or hormonal signals that might explain these patterns in legumes?
A: We find this question quite challenging, partly related to the very first one and generally going beyond the scope of our survey. Notably, some correlations are found in one or two groups out of three (or even in certain sections of a genus) additionally evidencing for the exceptional diversity of underlying mechanisms.

Q: Given the differences in reproductive traits between annual and perennial species, how might these findings inform breeding strategies for improving yield or stress resilience in legumes? Are there specific traits identified in your study that could be targeted for selection in breeding programs? If yes, it should be discussed properly.
A: As our work was not aimed at breeding or biotechnology, we would rather forbear from any practical recommendations. Considering how many species were taken into analysis on the pooled dataset, any sort of guidance is risky of becoming speculative and overgeneralizing. However, the most important outcome to be potentially implemented in breeding is the pronounced gap between the potential and real levels of productivity in perennial legumes. As the vast majority of legume crops are annuals with relatively high values of F/F and S/O ratios, there is a limited possibility to enhance their yield via the increase of these ratios by any measures. On the contrary, the yield of perennial legumes is promising for increasing. This has been emphasized in the Conclusion (lines 472-474).

Q: The seed-to-ovule and fruit-to-flower ratios differ between annual and perennial species. How do you interpret these differences in the context of reproductive success and fitness? Are these ratios consistent across different environments, or do they vary with ecological conditions?
A: These ratios may differ among years or populations of the same species; this phenomenon is now mentioned in lines 283-285. As for the interpretation of the difference between annuals and perennials in terms of F/F and S/O ratios, we suppose it is a sort of side effect of preferential outcrossing with the resulting gene load, inbreeding depression, and pollinator deficiency rather than a component of fitness. To overcome these difficulties, many more flowers and ovules are initiated which may hypothetically result in pods and seeds in some favourable conditions. This was added to the discussion (lines 327-330).

Q: The study notes a correlation between flower size and reproductive success. How might flower size influence pollinator behavior and effectiveness in annual versus perennial species? Could differences in pollinator interactions contribute to the observed patterns in reproductive traits?
A: We are not discussing this correlation as the available data does not provide enough information for this. Hypothetically, larger flowers are likely to be more attractive for animal pollinators due to the advantage in both visibility and often the amount of nectar produced. However, considering the manifold nature of reproductive success, it would be very difficult to isolate the contribution of floral size to the seed and fruit set. Disentangling this is also a complicated goal due to a variable degree of self-compatibility among species. Near isogenic lines of the same species differing in flower size might serve an interesting model for such a survey. In the absence of such a material, any conclusions may be premature. Moreover, the example of the Fabeae is notable as they have the pronounced difference in S/O and F/F ratios between annuals and perennials – but not in flower size. This was emphasized in the text (lines 220-221).

Q: Are there other traits, such as seed dormancy, germination timing, or root architecture, that you think could be important in understanding the life history strategies of annual versus perennial legumes?
A: These listed traits are probably a few in a long list of the features driven by lifespan. The exact cause-effect relationship is difficult to estimate in many cases. Our manuscript reports the results of the lasting investigation on lifespan-associated morphological adaptations in legumes. However, seed or root features, though important for the life history strategy, were not in the scope of our work presented in the paper.

Q: The study mentions a positive correlation between flower and seed sizes in Astragalus. Do you think this correlation is driven by genetic linkage, pleiotropy, or adaptive evolution? How would you design an experiment to test the causal relationship between these traits?
A: A positive correlation between flower and seed sizes is found both in Astragalus and the Fabeae (Table 1 in the manuscript) hence suggesting it is most likely shaped by adaptive evolution, especially taking into account the exceptionally high level of diversity in these taxa. However, the exact underlying regulatory mechanism may be diverse. Indeed, the increase of both flower and seed sizes may result from the pleiotropic action of some allele(s) or be governed by tightly linked genes, or some interaction between unlinked loci. To distinguish between these scenarios, either forward or reverse genetics approach should be applied which is hardly available now in Astragalus. It might be also helpful to perform crosses between lines with contrasting phenotypes or some induced mutagenesis studies. Nowadays, probably the easiest way is to apply transcriptomic analysis to reveal which genes are up- or downregulated in forms with contrasting flower/seed sizes. A question of mechanisms causing the covariation of sizes of different organs seems far from investigated in details and is worth becoming a subject of a special survey.

Thank you again for your interesting comments and questions which inspired us to revisit some parts of our work’s reasoning and logic. We hope that the changes we made will improve the readability and scientific soundness of our work.

Many thanks and kind regards,
Authors
